# Docetaxel-Cisplatin-Fluorouracil Induction Chemotherapy for Larynx Preservation in Patients with Locally Advanced Hypopharyngeal Cancer: Predictive Factors of Oncologic and Functional Outcomes

**DOI:** 10.3390/jcm12031131

**Published:** 2023-02-01

**Authors:** Pierre Mattei, Jocelyn Gal, Emmanuel Chamorey, Olivier Dassonville, Gilles Poissonnet, Déborah Aloi, Médéric Barret, Inga Safta, Esma Saada, Anne Sudaka, Dorian Culié, Alexandre Bozec

**Affiliations:** 1Institut Universitaire de la Face et du Cou, Centre Antoine Lacassagne, 33 Avenue de Valombrose, 06189 Nice, France; 2Department of Statistics, Centre Antoine Lacassagne, 33 Avenue de Valombrose, 06189 Nice, France; 3Department of Radiation Oncology, Centre Antoine Lacassagne, 33 Avenue de Valombrose, 06189 Nice, France; 4Department of Medical Oncology, Centre Antoine Lacassagne, 33 Avenue de Valombrose, 06189 Nice, France; 5Department of Pathology, Centre Antoine Lacassagne, 33 Avenue de Valombrose, 06189 Nice, France; 6Faculty of Medecine, Côte d’Azur University, 06107 Nice, France

**Keywords:** hypopharyngeal cancer, larynx preservation, induction chemotherapy, survival, dysphagia, predictive factors

## Abstract

Background: The aims of this study were to evaluate the clinical outcomes and their predictive factors in locally advanced hypopharyngeal cancer (HC) patients included in a docetaxel-cisplatin-fluorouracil induction chemotherapy (ICT)-based larynx preservation (LP) program. Methods: Between 2005 and 2021, 82 patients with a locally advanced resectable HC who received ICT in an LP program were included in this retrospective study. The predictors of oncologic and swallowing outcomes were determined in univariate and multivariate analyses. Results: The three- and five-year overall survival (OS) rates were 67 and 54%, respectively. The T4 tumor stage was the only predictive factor of poor response to ICT (*p* = 0.03). In multivariate analysis, a T stage = 4 (*p* = 0.02), an ICT cycle number < 3 (*p* = 0.003) and the absence of a response to ICT (*p* = 0.03) were significantly associated with worse OS. A low body mass index before therapy (*p* = 0.003) and enteral nutrition during therapy (*p* = 0.005) were significantly associated with severity of dysphagia 6 months after treatment. Conclusions: The T stage, number of ICT cycles performed and response to ICT are the main predictors of oncologic outcomes. Patients with T4 HC are poor candidates for LP and should be referred to immediate radical surgery.

## 1. Introduction

Hypopharyngeal carcinoma (HC) represents one of the most aggressive head and neck malignancies and is most often diagnosed at a locally advanced stage [1,2]. In most cases, patients with locally advanced HC are not amenable to conservative surgical procedures and, if operated on, require total laryngectomy (TL) with partial or circular pharyngectomy [2,3]. In order to avoid this radical surgery and its functional consequences, locally advanced HC patients are included, whenever possible, in a larynx preservation (LP) program consisting of either concurrent chemoradiation therapy (CRT) or induction chemotherapy (ICT) followed by RT (alone or with cisplatin or cetuximab) in good responders to ICT [2,4].

In France and Europe, the docetaxel-cisplatin-fluorouracil (TPF) ICT is the most popular strategy for LP in HC patients. Since most LP studies have only included laryngeal cancer patients or laryngeal and HC patients mixed together, the clinical results of such a therapeutic strategy are insufficiently explored in the specific context of HC patients [4].

The aims of this study were therefore to evaluate the oncologic and functional outcomes and their predictive factors in locally advanced HC patients included in a TPF ICT-based LP program.

## 2. Materials and Methods

### 2.1. Ethical Considerations

The study protocol was reviewed and approved by institutional ethics committees prior to the start of the study. Informed consent was obtained from each of the participants.

### 2.2. Subjects

Between January 2005 and June 2021, all previously untreated locally advanced (T3 or T4) HC patients amenable to radical surgery with TL, who received TPF-based ICT in an LP program, at our Institution were included in this retrospective study. The exclusion criteria were as follows: non-resectable disease, distant metastasis, recurrent tumor and patients unfit for receiving TPF-based ICT.

During the study period, 188 patients were treated for a locally advanced HC at our Institution. Thirty-three patients underwent primary TL followed by (C)RT, 45 patients who were unfit to receive TPF-ICT were treated by definitive concurrent CRT, 12 patients with an unresectable disease received (C)RT and 16 patients with distant metastasis at diagnosis were treated by systemic therapies. Therefore, a total of 82 patients were included in the study.

### 2.3. Induction Chemotherapy Protocol for Larynx Preservation

The therapeutic strategy was elaborated for each specific patient during a multidisciplinary tumor board (MTB) discussion.

Patients received two–three cycles of ICT with docetaxel (T: 75 mg/m^2^ on day 1), cisplatin (*p*: 75 mg/m^2^ on day 1) and fluorouracil (F: 750 mg/m^2^ per day on days 1 to 5) at 3-week intervals. Granulocyte colony-stimulating factor was administered prophylactically (one subcutaneous injection of pegfilgrastim per ICT cycle), but we did not use prophylactic antibiotics during ICT. Tumor response to ICT was evaluated two weeks after the last treatment cycle by endoscopic examination (nasofibroscopy or panendoscopy under general anesthesia) and head and neck CT-scan. ICT responders were defined by a reduction in primary tumor volume of at least 50% along with laryngeal mobility recovery, if initially impaired. The third cycle of ICT was optional and was used in cases of partial response to ICT to improve the response and only if the tolerance to ICT was satisfactory.

ICT responders received conventional external beam RT (one 2 Gy fraction per day, 5 days a week for a total of 70 Gy), with or without P (100 mg/m^2^ per day on days 1, 22 and 43 of RT) or cetuximab (E: loading dose 400 mg/m^2^ on day 1 of the week preceding RT and, thereafter, a weekly dose of 250 mg/m^2^ for the duration of RT).

Poor ICT responders were referred for TL with partial or circular pharyngectomy and bilateral neck dissection followed by adjuvant RT (one 2 Gy fraction per day, 5 days a week for a total of 60 to 66 Gy) or CRT (P: 100 mg/m^2^ per day on days 1, 22 and 43 of RT) in cases of positive surgical margins and/or extracapsular spread in metastatic cervical lymph nodes.

### 2.4. Follow-Up

Post-therapeutic follow-up was conducted in accordance with national guidelines. Patient clinical examination was scheduled every two months during the first two years, then every four months. A head and neck and thoracic CT-scan was performed 3 to 4 months after the initial treatment and once a year for the next 5 years or in cases of clinical suspicion of tumor recurrence.

### 2.5. Main Outcome Measures

Overall survival (OS), cause-specific survival (CSS) and recurrence-free survival (RFS) were determined by Kaplan–Meier analysis.

Patients’ general health status and comorbidity level were assessed using the American Society of Anesthesiologists (ASA) score.

Since swallowing disorders and oral diet modifications are among the main long-term sequelae of HC treatment, we evaluated the severity of dysphagia and the restriction of oral diet using the Dysphagia Outcome and Severity Scale (DOSS) [5]. This well-standardized seven-step scale is used to classify the severity of dysphagia, with level 7 corresponding to normal swallowing and level 1 to complete enteral feeding. Swallowing function was evaluated in patients who were still alive and disease-free 6 months after the end of the initial treatment.

### 2.6. Statistical Analyses

We analyzed the impacts on oncologic (response to ICT, risk of TL, OS, CSS and RFS) and functional (DOSS) outcomes of the following factors: age (< vs. ≥70 years), gender, ASA score (< vs. ≥3), T stage (T3 vs. T4), N stage (< vs. ≥2a), response to ICT, body mass index (BMI), weight loss before therapy (< vs. ≥10% of usual body weight), alcohol consumption and enteral nutrition during therapy.

Univariate analyses were performed using Log Rank tests (OS, CSS and RFS) or Chi-squared tests confirmed by Fisher’s exact tests (response to ICT, DOSS and feeding tube dependence). For multivariate analysis (conducted only when more than one factor was significant in univariate analyses), all variables associated with *p* < 0.05 in univariate analysis were included in Cox or logistic regression models with forward stepwise selection. All statistical analyses were performed at a 5% alpha risk or 95% confidence interval by the biostatistician using R.3.0.1 software on Windows.

## 3. Results

### 3.1. Sample Description

A total of 82 patients (68 men and 14 women, mean age: 58.6 ± 7.7 years, range: 30 to 79 years) were enrolled in this retrospective study. Their main clinical characteristics are shown in Table 1.

### 3.2. Response to Induction Chemotherapy

Three patients (3.7%) died during the ICT protocol after one (*n* = 2) or two cycles (*n* = 1) of TPF. Causes of death during ICT were as follows: febrile neutropenia (*n* = 1) and unknown (*n* = 2).

Among the 79 remaining patients, 58 received 3 cycles, 20 received 2 cycles and 1 received 1 cycle (severe toxidermia) of TPF. Sixty-five patients (82%) were considered ICT responders (including 29 complete responses) and received RT alone (*n* = 9) or associated with P (*n* = 41), carboplatin (*n* = 5) or cetuximab (*n* = 10).

Fourteen patients (18%) showed no significant response to ICT (tumor volume reduction < 50% or no recovery of larynx mobility), but only three of them underwent immediate radical surgery followed by adjuvant CRT. Six patients refused radical surgery and received CRT despite the absence of a response to ICT. Five patients did not recover larynx mobility after ICT despite a reduction in primary tumor volume > 50% and received RT with P (*n* = 3) or cetuximab (*n* = 2), since LP protocol continuation was considered acceptable after MTB discussion.

T4 tumor stage was the only predictive factor of a poor response to TPF ICT identified in the univariate analysis (*p* = 0.03). Indeed, the ICT response rate was 86.6% (58/67) and 58.3% (7/12) in patients with T3 and T4 tumor stages, respectively.

### 3.3. Oncologic Outcomes and Their Predictive Factors

The median follow-up was 75 months (95% CI: 49.1–117.7). The three and five-year OS, CSS and RFS rates were 67, 69 and 56% and 54, 61 and 50%, respectively. OS, CSS and RFS Kaplan–Meier survival curves are shown in Figure 1, Figure 2 and Figure 3.

Predictive factors of OS, CSS and RFS are presented in Table 2. Multivariate Cox regression analysis showed that a T stage = 4 (hazard ratio: HR: 2.71; 95%CI: 1.18–6.21; *p* = 0.02), ICT cycle number < 3 (HR: 2.76; 95%CI: 1.41–5.42; *p* = 0.003) and the absence of a response to ICT (HR: 2.46; 95%CI: 1.07–5.63; *p* = 0.03) were significantly associated with a worse OS. An age > 70 years (HR: 9.33; 95%CI: 1.75–49.74; *p* = 0.009), T stage = 4 (HR: 3.71; 95%CI: 1.46–9.42; *p* = 0.005) and ICT cycle number < 3 (HR: 2.90; 95%CI: 1.27–6.64; *p* = 0.01) were significantly associated with a worse CSS. A T stage = 4 (HR: 2.32; 95%CI: 1.08–4.97; *p* = 0.03), ICT cycle number < 3 (HR: 2.61; 95%CI: 1.38–4.95; *p* = 0.003) and enteral nutrition during therapy (HR: 1.98; 95%CI: 1.03–4.97; *p* = 0.04) were significantly associated with a worse RFS.

### 3.4. Larynx Preservation

Of the 82 patients included in the present study, 16 underwent radical surgery with TL due to an absence of a response to ICT (*n* = 3) or tumor persistence/recurrence after a complete LP protocol (ICT followed by RT alone or with P or cetuximab, *n* = 13).

Of the 11 patients who did not respond to ICT but pursued larynx preservation protocol, 7 patients displayed local recurrence in the follow-up period but only 3 of them could be managed by salvage TL.

After univariate and multivariate analysis, an ASA score ≥ 3 (Odds ratio: OR: 5.71; 95%CI: 1.44–23.56; *p* = 0.01) and the absence of a response to ICT (OR: 4.13; 95%CI: 1.07–16.03; *p* = 0.04) were significantly associated with a higher risk of TL.

### 3.5. Swallowing Outcomes

Dysphagia severity was evaluated in 78 patients (4 patients had died before the evaluation time) 6 months after therapy. The mean DOSS score was 5.58 ± 1.79. Seven patients (9%) had a DOSS score ≤ 2, indicating their dependence on enteral nutrition (enteral feeding alone: *n* = 5 or enteral feeding combined with partial oral nutrition: *n* = 2). The DOSS scores obtained for the 78 evaluable patients 6 months after therapy are presented in Table 3.

After univariate and multivariate analysis, a low BMI before therapy (OR: 6.78; 95%CI: 2.07–26.43; *p* = 0.003) and enteral nutrition during therapy (OR: 5.52; 95%CI: 1.75–19.25; *p* = 0.005) were significantly associated with a worse DOSS score (DOSS < 6) 6 months after therapy. The other clinical factors (age, T and N stage, and response to ICT, etc.) tested in the study had no significant impact on the post-treatment DOSS score.

## 4. Discussion

Since most LP studies have only included laryngeal cancer patients or laryngeal cancer and HC patients mixed together, there are few data on the clinical outcomes of patients with locally advanced HC included in an LP program [4]. Moreover, disappointing results have been reported following the transfer into clinical practice of LP protocols [6,7]. Indeed, in the United States, a decreased survival was observed for patients with laryngeal cancer in the mid-1990s, potentially related to changes in management patterns, with an increased use of chemoradiation along with a decreased use of radical surgery [6]. The lessons learned from the use of LP protocols in common clinical practice have shown that LP strategies have been used for patients with more advanced tumors than those included in original LP studies [4,6,7]. This highlights the fundamental role of the appropriate selection of patient candidates for an LP therapeutic approach. Knowledge of the potential predictive factors of oncologic and functional outcomes following an ICT-based LP program in locally advanced HC patients is therefore of critical importance to guide patient selection.

In the present study, we showed an 82% response rate to TPF ICT (tumor volume reduction > 50% and recovery of larynx mobility if initially impaired), with 58 patients (71%) receiving three cycles of TPF and 3 patients who died during the ICT protocol (4%). Similar results were reported by Lefebvre et al. in the TREMPLIN phase II randomized LP study that enrolled 59% of patients with locally advanced HC (ICT response rate = 85%, 74% of patients undergoing three cycles of ICT, 3% of mortality during ICT) [8]. Comparable results have also been reported by Pointreau et al. in a phase III randomized trial comparing TPF with PF ICT for LP [9]. In this study, that included 54% of HC patients, the overall response rate after ICT was 80.0% in the TPF group (41.8% complete response and 38.2% partial response) and 59.2% in the PF group (30.1% complete response and 29.1% partial response); (difference = 20.8%; *p* = 0.002). In the TPF group, three patients (2.7%) died during the ICT protocol [9].

With 3 and 5-year OS rates of 67 and 54%, respectively, the oncologic outcomes of the present study appear to be satisfactory for patients with locally advanced HC. In the study from Pointreau et al. mentioned above, a 3-year OS rate of 60% was reported in both treatment arms (TPF and PF) [9]. The long-term results of this study showed, in the TPF group, OS rates of 51% and 30%, at 5 and 10 years, respectively [10]. In the TREMPLIN study, the 5-year OS rate was 67% [8]. That is greater than our results, but this study enrolled not only locally advanced HC patients but also patients with laryngeal cancer (41%) and less advanced disease (T2: 11%) [8]. In a recent retrospective analysis of 142 consecutive patients with locally advanced HC receiving different types of ICT regimens for LP, Li et al. found a 3-year OS rate of 54% [11]. Altogether, these results indicate that, in clinical practice, a TPF ICT-based LP approach represents an adequate therapeutic option for locally advanced HC patients.

Interestingly, the present study showed that the T4 tumor stage was associated with poor ICT response rates and worse oncologic outcomes. Indeed, the T4 tumor stage was an independent predictive factor for both OS, CSS and RFS. Although T4 primary tumors are known to be poor candidates for LP strategies in laryngeal cancer patients, such an association had not been adequately demonstrated for HC patients [7]. In a retrospective analysis of 616 patients with T4a laryngeal cancer treated in the United States between 2003 and 2006, Grover et al. showed that most patients received CRT for LP, despite guidelines suggesting TL as the preferred initial approach [12]. Patients receiving CRT for LP had worse OS compared with those undergoing TL. The authors concluded that previous studies of non-T4a locally advanced larynx cancer showing no difference in survival between CRT and TL may not apply to T4a disease and that patients should be counseled accordingly [12]. Similarly, the present study identified T4a HC patients as poor candidates for the LP therapeutic strategy, which suggests that patients with T4a HC should be referred to immediate radical surgery.

Other independent predictive factors of poor oncologic outcomes found in the present study were: an ICT cycle number < 3 (for OS, CSS and RFS), the absence of a response to ICT (for OS), an age > 70 years (for CSS) and enteral nutrition during therapy (for RFS). In addition to the obvious decrease in therapeutic intensity, the correlation between an incomplete ICT protocol and patient prognosis may be a reflection of greater patient vulnerability, explaining a poor tolerance to TPF ICT. The absence of a response to ICT is traditionally an indication for immediate radical surgery. However, in clinical practice, a significant proportion of patients refuse TL, and the choice between continuation of LP strategy or switching to radical surgery is debatable since the criteria retained to define an objective response may be incompletely obtained (a reduction in tumor volume close to but <50%, a reduction in tumor volume > 50% but without recovery of larynx mobility, etc.). Indeed, 11 out of the 14 patients who did not reach an objective response to ICT pursued the LP protocol and seven of them developed local recurrence in the follow-up period. This result suggests strongly recommending radical surgery with TL in cases of a poor response to ICT and providing exhaustive and appropriate information to these patients. The impact of enteral nutrition during therapy on RFS may be explained by a poor tolerance to treatment, with this being potentially responsible for treatment interruption/discontinuation or intensity reduction and possibly reflecting a more aggressive disease and/or higher patient vulnerability.

An ASA score ≥ 3 and the absence of a response to ICT were associated with a decreased rate of larynx preservation. This primarily suggests only including in TPF ICT-based LP protocol patients with a good general health status and a low comorbidity level. This type of selection criteria was essential in original LP studies and should also be respected when applying this therapeutic strategy in clinical practice [4,7].

With a mean DOSS score of 5.58 and 69% of patients recovering a normal diet, the swallowing outcomes of the patients included in the present study were satisfactory. The rate of patients dependent on enteral nutrition (DOSS score ≤ 2) is important to consider because it reflects a failure in functional LP and represents an objective criterion of comparison available in most studies, even retrospective ones. The 9% enteral feeding rate reported in our study is relatively low compared with literature data. Indeed, compared with other head and neck malignancies, HC has been associated with a higher risk of feeding tube dependence [13,14]. In a retrospective review of 43 HC patients treated with nonsurgical modalities (RT or CRT), Bhayani et al. reported a rate of feeding tube dependence 1 year after therapy of 28%, with a feeding tube dependence duration significantly shorter in those who maintained oral intake at the end of treatment [14]. This data is in line with the results of the present study, which show that dysphagia requiring enteral nutrition during therapy was a significant predictive factor of persistent dysphagia 6 months after the end of the treatment. This indicates that most patients exhibiting severe swallowing impairments during therapy will not recover a satisfactory swallowing function after treatment. In this regard, for most authors, patients with poor laryngoesophageal function at baseline are considered poor candidates for LP [4]. Locally advanced HC patients are at high risk of dysphagia and nutritional problems at all stages of their disease management [15,16]. In a recent retrospective study of 310 head and neck cancer patients treated with (C)RT, Bojaxhiu et al. demonstrated that patients aged > 60 years with HC, tobacco consumption and poor performance status displayed a higher risk of gastrostomy tube-related complications leading to an unplanned hospitalization [16]. As shown in the present study, a low BMI at baseline is associated with a higher level of dysphagia after therapy, highlighting the importance of the comprehensive management of nutritional problems throughout the multimodal management of these patients.

The main limitations of this study lie in its retrospective and monocentric nature. It nevertheless represents a comprehensive description of the clinical outcomes of a relatively large series of consecutive patients, all treated in a TPF ICT-based LP program for locally advanced HC. To our knowledge, this is the first study analyzing the predictive factors of oncologic and swallowing outcomes in this specific population. We believe that the results presented above are of paramount importance to refine the selection of and information regarding patients, and, ultimately, to improve the global management of HC patients.

## 5. Conclusions

TPF ICT-based LP is a reliable therapeutic option for patients with locally advanced HC, providing high tumor response rates and satisfactory oncologic outcomes, with a 5-year OS above 50%. The T stage, number of ICT cycles performed and response to ICT are the main predictors of oncologic outcomes. Since the T4 tumor stage is associated with the absence of a response to ICT and poor survival rates, patients with T4 HC are poor candidates for LP and should be referred to immediate radical surgery. The absence of a response to ICT is associated with a high risk of treatment failure when the LP protocol is continued despite a poor response to ICT. A low BMI before therapy and the requirement of enteral nutrition during therapy are predictors of decreased swallowing function after treatment.

## Figures and Tables

**Figure 1 jcm-12-01131-f001:**
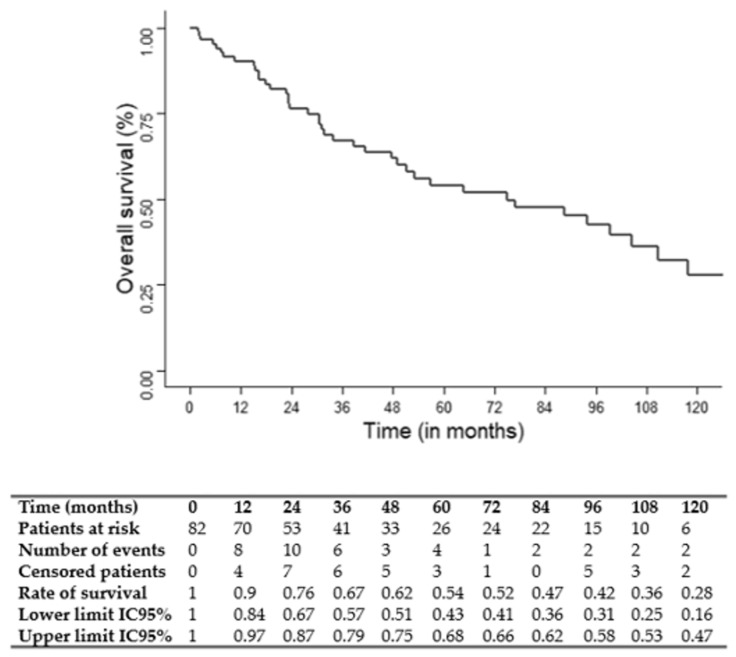
Kaplan–Meier overall survival (OS) curve.

**Figure 2 jcm-12-01131-f002:**
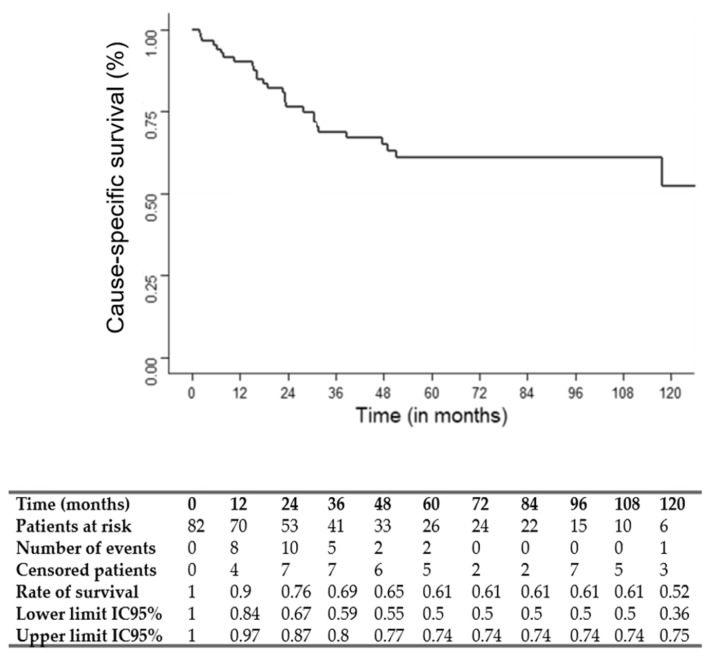
Kaplan–Meier cause-specific survival (CSS) curve.

**Figure 3 jcm-12-01131-f003:**
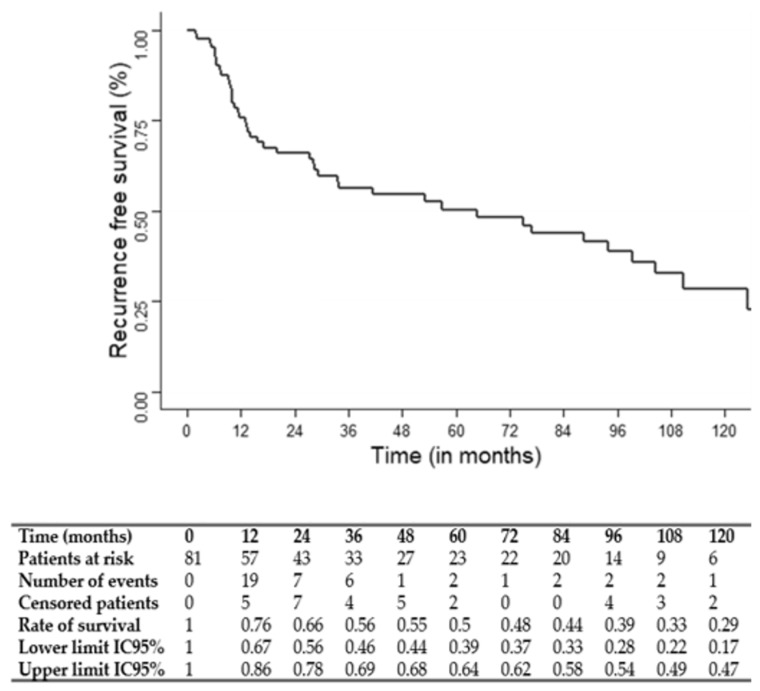
Kaplan–Meier recurrence-free survival (RFS) curve.

**Table 1 jcm-12-01131-t001:** Patients’ clinical characteristics.

Clinical Characteristics	*n* = 82 (%)
Gender: male/female	68 (83)/14 (17)
Age: </≥70 years	78 (95)/4 (5)
ASA score: </≥3	70 (85)/12 (15)
BMI: <18/18–20/>20	5 (6)/9 (11)/68 (83)
Weight loss before therapy: </≥10% of UBW	60 (73)/22 (27)
Alcohol abuse: yes/no	44 (54)/38 (46)
Tobacco consumption: </≥10 PY	8 (10)/74 (90)
T stage: T3/T4	69 (84)/13 (16)
N stage: 0/1/2a-c/3	16 (20)/13 (16)/43 (52)/10 (12)

ASA, American Society of Anesthesiologists; BMI: body mass index; UBW: usual body weight; PY: pack years.

**Table 2 jcm-12-01131-t002:** Predictive factors of oncologic outcomes.

Predictive Factors	OS (UA/MA)	CSS (UA/MA)	RFS (UA/MA)
Gender: male/female	NS	NS	NS
Age: </≥70 years	0.02/0.12	0.01/0.009	NS
ASA score: </≥3	NS	NS	NS
BMI: </≥18	NS	NS	NS
Weight loss before therapy: </≥10% UBW	NS	NS	NS
Alcohol consumption: yes/no	NS	NS	NS
T stage: T3/T4	0.006/0.02	<0.001/0.005	0.002/0.03
N stage: 0/1–3	NS	NS	NS
N stage: 0–1/2a-3	NS	NS	NS
ICT cycle number: 1–2/3	0.001/0.003	0.001/0.01	0.003/0.003
Response to ICT: yes/no	0.01/0.03	0.004/0.07	0.008/0.06
Enteral nutrition during therapy: yes/no	0.04/0.26	NS	0.02/0.04

OS: overall survival; CSS: cause-specific survival; RFS: recurrence-free survival; ASA: American Society of Anesthesiologists, BMI: body mass index, UBW: usual body weight, ICT: induction chemotherapy; *p* values in univariate analyses (UA) using Log rank tests and in multivariate analyses (MA) using Cox regression models, statistically significant *p* values (<0.05) are underscored; NS: non statistically significant (*p* > 0.05).

**Table 3 jcm-12-01131-t003:** Dysphagia outcome and severity scale (DOSS) score 6 months after therapy.

DOSS Score	Number of Patients (*n* = 78)	Percentage (%)
**Full per-oral nutrition and normal diet**		
-7: normal diet in all situations	32	41
-6: normal diet with functional limits/modified independence	22	28
**Modified diet and/or independence**		
-5: mild dysphagia	7	9
-4: mild to moderate dysphagia	5	6
-3: moderate dysphagia	5	6
**Enteral nutrition necessary**		
-2: moderate to severe dysphagia, partial per-oral nutrition only	2	3
-1: severe dysphagia: unable to tolerate any per-oral nutrition	5	6

## Data Availability

The data presented in this study are available on request from the corresponding author. The data are not publicly available due to privacy.

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
