# Peer review of "Docetaxel-Cisplatin-Fluorouracil Induction Chemotherapy for Larynx Preservation in Patients with Locally Advanced Hypopharyngeal Cancer: Predictive Factors of Oncologic and Functional Outcomes"

_jcm, 2023, doi:10.3390/jcm12031131_

Round 1
Reviewer 1 Report
Well done on a well executed paper. In page 2 I would say exclusion rather then non- inclusion criteria
Reviewer 2 Report
Mattei et al., evaluated the clinical outcomes and predictive factors in locally advanced HC patients included in ICT based LP program. Based on the data from 82 patients, they conclude tumor stage, ICT cycles and response to ICT as main predictors of oncologic outcomes. They also recommend that the patients with T4 HC are poor candidates for LP and therefore referred to immediate surgery. In the introduction, they mention that previous studies examined mostly laryngeal or laryngeal/HC patients and therefore they evaluated the patient outcome in the context of HC patients only. The conclusions and interpretations of the study are clear, and I recommend the publication of this article in JCM after addressing the following points.
Major Concern:
1. Line 291
The authors state that enteral nutrition during therapy is significantly associated with a worse DOSS score and is a predictive factor for persistent dysphagia.
Here, the patients who had enteral nutrition during treatment is significantly associated to have a worse DOSS score after treatment.
a. The statement has to be carefully reworded as the dependance on enteral nutrition is already indicative of bad DOSS scores and in turn severe dysphagia; and post treatment, the patient still has bad DOSS scores. This association does not help provide any relevant information for the counselling treatment options to patients.
b. The study does not assess whether the absence of response to ICT treatment affects the DOSS score post treatment. If there is any data available in the collected database, please go ahead and mention it here in the article.
The authors also say that enteral feeding is a predictive factor for persistent dysphagia. The use of the term ‘predictive factor’ implies the association of a future event with some current status. Doesn’t enteral feeding already imply dysphagia?
A better approach would be to assess the changes in DOSS score (before and after the ICT treatment) in order to better understand if low DOSS scores are a “predictive outcome” for having continued enteral nutrition post treatment.
2. Table 1 : Clinical Characteristics.
This is not a concern but mostly a suggestion. In the table, I noticed that the Alcohol intake of the patients is almost evenly divided. 44 vs 38 for yes/no. Would it be interesting to take these two sets of patients and look at the response to ICT, survival characteristics etc.?
It could be an additional figure/data table or can be incorporated into the existing table 2?
If there is any significance between the two groups in any of the outcomes, that will be a better insight for the clinicians to decide on the treatment options?
Minor Concerns:
1) Please include the number of patients in the Abstract under Methods so that it is easy for readers to understand the significance of the study. “Between 2005 and 2021, all….. (here instead of all, please indicate the number). Alternatively, a separate sentence can be added too.
2) In the “Institutional review board statement”, the protocol code is missing. It is merely mentioned XXX.
Reviewer 3 Report
The manuscript is a single institution review of a series of patients with hypopharyngeal squamous cell carcinoma treated with induction TPF chemotherapy followed by concurrent C/RT. As such, the number of patients is good, and uniformity treatment also good. It is well written. Some patients refuse surgery if they do not achieve an adequate response to induction, and the authors also report on the outcome of these patients, a valuable addition. Patients with unresectable disease were excluded from the series.
The number of patients in the series should be mentioned in the abstract, and in section "2.2 Subjects".
The abstract should note that population studied is "locally advanced resectable HC patients"
Presumably many of the patients treated during that timespan (2005-2021) had unresectable disease. They would not be candidates for the same "protocol", having no option of diversion to surgery, but it would be of interest if the authors could report on the number of such cases and the outcome.
Reviewer 4 Report
The aims of the study were to evaluate the oncologic and functional outcomes and their predictive factors in locally advanced HC patients included in a TPF ICT- based LP program. The authors found that TPF ICT-based LP is a reliable therapeutic option for patients with locally advanced HC, providing high tumor response rates and satisfactory oncologic outcomes, with a 5 year OS above 50%. T stage, number of ICT cycles performed and response to ICT are the main predictors of oncologic outcomes. Since T4 tumor stage is associated with an absence of response to ICT and poor survival rates, patients with T4 HC are poor candidates for LP and should be referred to immediate radical surgery. The absence of response to ICT is associated with a high risk of treatment failure when the LP protocol is continued de- spite the poor response to ICT. Low BMI before therapy and requirement for enteral nutrition during therapy are predictors of decreased swallowing function after treatment.
The article is well described and the results are interesting.
Round 2
Reviewer 2 Report
The authors addressed all the comments satisfactorily. I recommend the manuscript to be published in the journal.